# Comparative Transcriptomics Identify Key Hypothalamic Circular RNAs that Participate in Sheep (*Ovis aries*) Reproduction

**DOI:** 10.3390/ani9080557

**Published:** 2019-08-14

**Authors:** Zhuangbiao Zhang, Jishun Tang, Xiaoyun He, Mingxia Zhu, Shangquan Gan, Xiaofei Guo, Xiaosheng Zhang, Jinlong Zhang, Wenping Hu, Mingxing Chu

**Affiliations:** 1Key Laboratory of Animal Genetics and Breeding and Reproduction of Ministry of Agriculture and Rural Affairs, Institute of Animal Science, Chinese Academy of Agricultural Sciences, Beijing 100193, China; 2Institute of Animal Husbandry and Veterinary Medicine, Anhui Academy of Agricultural Sciences, Hefei 230031, China; 3Agricultural College, Liaocheng University, Liaocheng 252059, China; 4State Key Laboratory of Sheep Genetic Improvement and Healthy Production, Xinjiang Academy of Agricultural and Reclamation Sciences, Shihezi 832000, China; 5Tianjin Institute of Animal Sciences, Tianjin Academy of Agricultural Sciences, Tianjin 300381, China

**Keywords:** circular RNA, hypothalamus, GnRH, reproduction, sheep

## Abstract

**Simple Summary:**

The hypothalamus plays crucial roles in sheep reproduction. However, the expression profiles of sheep hypothalamic circular RNA (circRNA), which has been proved to exert important functions in many physiological processes, remain largely unknown. In this study, we used RNA sequencing to explore the expression of circRNAs in the hypothalamus of sheep with the *FecB ++* genotype. The results suggested that several key hypothalamic circRNAs may participate in sheep reproduction by influencing gonadotropin-releasing hormone (GnRH) activities or affecting key gene expression indirectly or directly. This study provides a further reference for understanding the differences of sheep fecundity.

**Abstract:**

Circular RNA (circRNA), as an emerging class of noncoding RNA, has been found to play key roles in many biological processes. However, its expression profile in the hypothalamus, a powerful organ initiating the reproductive process, has not yet been explored. Therefore, we used RNA sequencing to explore the expression of circRNAs in the hypothalamus of sheep with the *FecB* ++ genotype. We totally identified 41,863 circRNAs from sheep hypothalamus, in which 333 (162 were upregulated, while 171 were downregulated) were differentially expressed in polytocous sheep in the follicular phase versus monotocous sheep in the follicular phase (PF vs. MF), moreover, 340 circRNAs (163 were upregulated, while 177 were downregulated) were differentially expressed in polytocous sheep in the luteal phase versus monotocous sheep in the luteal sheep (PL vs. ML). We also identified several key circRNAs including oar_circ_0018794, oar_circ_0008291, oar_circ_0015119, oar_circ_0012801, oar_circ_0010234, and oar_circ_0013788 through functional enrichment analysis and oar_circ_0012110 through a competing endogenous RNA network, most of which may participate in reproduction by influencing gonadotropin-releasing hormone (GnRH) activities or affecting key gene expression, indirectly or directly. Our study explored the overall expression profile of circRNAs in sheep hypothalamus, which potentially provides an alternative insight into the mechanism of sheep prolificacy without the effects of *FecB* mutation.

## 1. Introduction

Sheep is an important agricultural species, the reproductive traits of which are strictly controlled by reproduction-related hormones, such as gonadotropin-releasing hormone (GnRH), follicle-stimulating hormone (FSH), and luteinizing hormone (LH) [1]. In this species, reproductive activities are initiated by GnRH pulsatile secretion, which reaches the pituitary, causing the release of FSH and LH. Then, those two key hormones act on the ovary, leading to the development of follicles and ovulation. GnRH, as an activator of female reproduction, is particularly important in this context, and both the amplitude and the frequency of GnRH pulsatile release have major effects on the female estrous cycle [2]. For example, higher pulse frequencies are associated with the selective secretion of LH in the consistent presence of FSH [3]. In addition, different GnRH pulsatility can activate different downstream genes. For example, increased LHβ was found to respond to rapid GnRH pulsatility, while the production of FSHβ was promoted in response to lower GnRH pulsatility [4]. Therefore, the hypothalamus, as a GnRH-generating organ, plays indispensable roles in female reproduction.

FecB was the first mutation identified in the bone morphogenetic protein receptor IB (*BMPRIB*) gene (A746G) and shown to be significantly associated with sheep reproduction. *FecB* has been found in many sheep breeds [5,6,7], including Small Tail Han sheep. One copy of the *FecB* mutation can significantly increase little size by 1, and two copies can increase by 1.5 [8]. Therefore, sheep can be categorized into those having the *FecB* BB genotype (two copies of *FecB* mutation), the *FecB* B+ genotype (one copy of *FecB* mutation), or the *FecB* ++ genotype (without *FecB* mutation). Sheep with *FecB* ++ normally have one offspring, but cases of such sheep having two or even three offspring have also been observed [7]. The mechanism underlying this remains to be explored.

All hormones such as GnRH, FSH, and LH are proteins and are mainly regulated at the transcriptional and post-transcriptional levels during their synthesis and release [9]. Therefore, diverse noncoding RNAs should also be considered to play key roles in modulating the reproductive process. With regard to the pituitary, researches have revealed that noncoding RNAs such as microRNAs [10,11], long noncoding RNAs [12], and the emerging class of circular RNAs (circRNAs) [9] are involved in reproductive modulation. CircRNAs are produced in eukaryotes by precursor mRNA through the back-splicing of exons. Their known functions include acting as sponges of cytoplasmic microRNAs to modulate gene expression, sequestering proteins, and sometimes being translated into polypeptides [13]. Regarding the hypothalamus, some researches have investigated the mechanism by which GnRH controls reproduction. Our previous work demonstrated that the long non-coding RNA MSTRG.26777, MSTRG.105228, and MSTRG.95128 may play important roles in the hypothalamus of sheep with *FecB* ++ by affecting GnRH activities indirectly or directly [14]. In addition, miRNA-200/429 and miRNA-155, as major components of switch, were found to regulate the production of GnRH by targeting *Zeb1* and *Cebpb*, two repressors of GnRH activating [15]. However, little is known about the expression profile of circRNAs in the hypothalamus and their involvement in the activities of reproductive hormones.

Thus, this study tries to explore the expression profile of circRNAs in the hypothalamus of sheep with the *FecB* ++ genotype in an effort to identify key circRNAs involved in the reproductive process, as well as reveal their potential functional mechanisms. This work is expected to provide alternative insights into the mechanism of sheep prolificacy in the hypothalamus.

## 2. Materials and Methods

### 2.1. Animal Processing

All of the animals involved in this study were approved by the Science Research Department (in charge of animal welfare issues) of the Institute of Animal Sciences, Chinese Academy of Agricultural Sciences (IAS-CAAS) (Beijing, China), and ethical approval was given by the Animal Ethics Committee of the IAS-CAAS (No. IASCAAS-AE-03).

First of all, the TaqMan probe method [16] was used to genotype the herds (n = 890) of Small Tail Han sheep (STH sheep), and the identified the STH sheep with the *FecB* ++ genotype were then divided into a monotocous group (n = 6, litter size ≥ 2) and a polytocous group (n = 6, litter size = 1) based on litter size records (the details of litter size have provided in our previous report [14]). All of the sheep involved in this study were bred at the sheep farm of Tianjin Institute of Animal Sciences under the same conditions and with free access to water and feed.

All of the selected STH sheep were treated with CIDR (controlled internal drug releasing; Zoetis Australia Pty., Ltd., NSW, Australia; progesterone 300 mg) for 12 days. Six sheep including three monotocous and three polytocous ones were slaughtered within 45–48 h of CIDR removal (follicular phase); the remaining six sheep were slaughtered in the same way on day 9 after CIDR removal (luteal phase). Therefore, all of the STH sheep were categorized as polytocous sheep in the follicular phase (PF), polytocous sheep in the luteal phase (PL), monotocous sheep in the follicular phase (MF), or monotocous sheep in the luteal phase (ML).

### 2.2. Tissue Acquisition, RNA Extraction and Sequencing

The tissue of the whole hypothalamus was obtained immediately after sheep slaughter and stored in liquid nitrogen (−80 °C) for RNA extraction. Total RNA was isolated from the 12 hypothalamic samples using the TRIzol Reagent (Invitrogen, Carlsbad, CA, USA) in accordance with the manufacturer’s instructions. To obtain high-quality RNA, 1% agarose electrophoresis and an Agilent 2100 Bioanalyzer (Agilent Technologies, Santa Clara, CA, USA) were used to examine the integrity and concentration of the extracted RNA. The purity of isolated RNA was also ensured using an Agilent RNA 6000 Nano Kit (Agilent Technologies).

All the sequencing data were obtained from Annoroad Gene Technology Co., Ltd. (Beijing, China). Samples of 3 μg of total RNA from each hypothalamus were pooled to construct the RNA library. First, ribosomal RNA was removed using the Ribo-Zero™ Gold Kit (Epicentre, Madison, WI, USA). Then, a fragmentation buffer was added to break the RNA into fragments, which were used as templates to synthesize the first strand of complementary DNA. The second strand of cDNA was also synthesized in the presence of dNTPs, ribonuclease H, and DNA polymerase I. The obtained double-stranded cDNA was processed with end-repair, the addition of base A and sequencing adaptors, and Uracil-N-Glycosylase (UNG) enzyme digestion. Subsequently, the polymerase chain reaction was performed to construct an RNA library. Finally, the desired RNA fragments were selected for sequencing (Illumina HiSeq 2500).

### 2.3. Data Quality Control and Circular RNA Identification

The clean reads were obtained from the raw reads after removing the reads with adaptor contamination (i.e., reads with more than five contaminated bases), low-quality reads (i.e., reads in which more than 15% of the total bases had a mass value Q ≤ 19), and reads with a rate of N (i.e., the rate of bases not recognized in a read) greater than 5%.

First, the reference genome and annotation files were downloaded from ENSEMBL (http://www.ensembl.org/index.html). Then, the BWA-MEM algorithm, which can align reads to the genome rapidly and efficiently and also supports the mapping segmental alignment of sequences to genomes, was applied to map clean reads to the *Ovis aries* reference genome (Oar v.3.1). Subsequently, a “CircRNA Identifier (CIRI)” strategy [17] was conducted to identify circRNAs, which included the following main steps: The use of the BWA-MEM algorithm to split sequence and then alignment, which was followed by scanning of the aligned sequence alignment/map (SAM) file to find the paired chiastic clipping paired-end mapping and GT-AG splicing signal. Finally, the sequence containing a junction site was re-mapped to ensure the reliability of the identified circRNAs.

### 2.4. Analysis of Differentially Expressed circRNAs and Functional Enrichment Analysis

To better describe the expression level of circRNAs, SRPBM (spliced reads per billion mapping) [18] was used to represent the expression of circRNAs. DESeq was then conducted to detect the differentially expressed circRNAs (DE circRNAs) [19]. For the purpose of screening key circRNAs, the thresholds of fold change > 1.5 and *p* < 0.05 were set to identify DE circRNAs.

We performed Gene ontology and Kyoto Encyclopedia of Genes and Genomes analyses using host genes of circRNA due to a lack of circRNA annotation. A particular GO term or KEGG pathway with *p* < 0.05, which was determined by the hypergeometric test method, was considered to reflect significant enrichment.

### 2.5. Integral DE circRNA–miRNA Network Analysis and ceRNA Construction

Many studies have reported that circRNAs could function as sponges of miRNAs to modulate gene expression [20,21]. To explore the potential interactome of circRNAs and miRNAs, the miRanda database (http://www.mirbase.org/index.shtml) was searched to identify sites on DE circRNAs to which miRNAs bind to build a circRNA-miRNA interactome using Cytoscape software [22]. We also constructed a competing endogenous RNA (ceRNA) network involving oar_circ_0012110 by searching for target genes of oar-miR-665-3p in the TargetScan database (http://www.targetscan.org/vert_72/).

### 2.6. Data Validation

To examine the accuracy of RNA sequencing, the divergent primers of six circRNAs (oar_circ_0012110, oar_circ_0022458, oar_circ_0029952, oar_circ_0033078, oar_circ_0025689, oar_circ_0030289) were synthesized by Beijing Tianyi Huiyuan Biotechnology Co., Ltd. (Beijing, China) (Appendix A). Then, reverse transcription was performed using PrimeScript™ RT reagent kit (TaKaRa, dalian, China), followed by the use of SYBR Green qPCR mix (TaKaRa, dalian, China) to conduct real-time quantitative polymerase chain reaction (RT-qPCR) through the Roche Light Cycler^®^480II system (Roche Applied Science, Mannheim, Germany). Finally, the data obtained from RT-qPCR were calculated with the normalization of β-actin (the details of the calculation method have been reported previously [14]).

## 3. Results

### 3.1. Circular RNA Expression Profiling

We obtained a total of 1,460,254,556 raw reads after sequencing; the mapped reads numbered 1,459,435,298, so the mapping rate in each sample reached nearly 100% (Appendix A). We identified 41,863 circRNAs in total from 12 hypothalamic samples (Appendix A), most of which are distributed on chromosome 1, followed by chromosomes 2 and 3 (Figure 1). Diverse types of circRNAs were also identified, the majority of which were classical circRNAs. In addition, the four sheep groups, namely PF (Figure 2A), PL (Figure 2B), MF (Figure 2C), and ML (Figure 2D), exhibited similar circRNA types and percentages. To obtain a better understanding of the characteristics of the circRNAs, we also examined the exon number and length in PF, PL, MF, and ML. As Figure 4 shows, most circRNAs in PF (Figure 3A), PL (Figure 3C), MF (Figure 3E), and ML (Figure 3G) had three or four exons; furthermore, the exon length of the circRNAs containing only one exon in PF (Figure 3B), PL (Figure 3D), MF (Figure 3F), and ML (Figure 3H) was greater than that of circRNAs having more than one exon.

### 3.2. Identification of Differentially Expressed Circular RNAs (DE circRNAs) and Functional Enrichment Analysis

To screen the key circRNAs, we set the thresholds of fold change > 1.5 and *p* < 0.05 to identify DE circRNAs. In total, we identified 333 DE circRNAs in polytocous sheep in the follicular phase versus monotocous sheep in the follicular phase (PF vs. MF), where 162 were upregulated, while 171 were downregulated (Figure 4A, Appendix A). We also identified 340 DE circRNAs in polytocous sheep in the luteal phase versus monotocous sheep in the luteal phase (PL vs. ML), where 163 were upregulated, while 177 were downregulated (Figure 4B, Appendix A). The heat maps of DE circRNAs in PF, MF, PL, and ML indicated the differences in expression pattern between PF and MF (Figure 4C) and between PL and ML (Figure 4D). We also conducted RT-qPCR to confirm the reliability of our sequencing data. The results demonstrated that the six selected circRNAs displayed expression trends similar to those in the sequencing results, indicating the accuracy of our sequencing data (Figure 5).

To better understand the functions of DE circRNAs, we conducted GO term and KEGG pathway analyses. The GO enrichment analysis showed that the top enriched GO terms in PF vs. MF were vesicle-mediated transport in the main category of biological process, phosphatidylinositol-3,5-bisphosphate 5-phosphatase activity in molecular function, and cytosol in the cellular component (Figure 6A, Appendix A). Meanwhile, the top enriched GO terms in PL vs. ML were cellular component organization in the main category of biological process, anion binding in molecular function, and neuron part in the cellular component (Figure 6B, Appendix A). KEGG enrichment analysis indicated that the most enriched pathway in PF vs. MF was morphine addiction; the transforming growth factor-β signaling pathway associated with reproduction was also enriched (Figure 7A, Appendix A). Regarding PL vs. ML, the most enriched pathway was endocytosis. Some pathways associated with reproduction including oxytocin signaling pathway were also enriched (Figure 7B, Appendix A).

### 3.3. Integral circRNA-miRNA Pairs and Competing Endogenous RNAs Analysis

We constructed circRNA-miRNA interactive networks to explore the potential functions of the DE circRNAs. We selected the top ten down- and up-DE circRNAs to construct a circRNA-miRNA interactome for both PF vs. MF and PL vs. ML (Figure 8, Appendix A). In total, 20 circRNA-miRNA pairs were constructed in each of PF vs. MF and PL vs. ML. Two important circRNA-miRNA pairs, oar_circ_0012110 targeted by oar-miR-665-3p and oar_circ_0033078 targeted by oar-miR-410-5p, were identified, among which oar_circ_0012110 and oar_circ_0033078 showed the largest fold change in PF vs. MF and PL vs. ML, respectively.

We also constructed ceRNA networks involving oar_circ_0012110. We searched the target gene of oar-miR-665-3p from the TargetScan database and selected the top ten target genes according to the binding score to construct a ceRNA network (Figure 9, Appendix A). However, we failed to predict the target genes of oar_circ_0033078 from the TargetScan database due to the poor conservation of oar_circ_0033078 in animal species.

## 4. Discussion

The hypothalamus, as a key brain region initiating reproductive activities, can generate a GnRH signal to modulate the secretion of downstream hormones such as FSH and LH. The production of GnRH is co-regulated by many factors, including kisspeptin, estrogen, and progesterone [23], and some metabolic activities of leptin, insulin, and ghrelin [24,25]. Accumulating research has focused on the mechanism of GnRH generation, but the corresponding genetic mechanism is yet to be understood [26,27]. Recent advances on circRNAs in sheep pituitary have provided new insights into the complexity of reproduction [9]. Considering the lack of detailed circRNAs expression profiles in sheep hypothalamus, determining the expression profiles of circRNAs and their potential functions in the hypothalamus may also deepen the understanding of sheep reproduction. Within the mammalian brain, the hypothalamus is a region that is highly enriched with circRNAs and in which most of the circRNAs are highly conserved [28]. In this key endocrine organ, we found 41,863 circRNAs, but some of them such as oar_circ_0033078 were poorly conserved. This indicates that circRNAs in the hypothalamus may execute diverse functions in different functional regions, such as the arcuate nucleus and the preoptic area. In addition, previous work also demonstrated the existence of 12,468 circRNAs in sheep pituitary, which were one-third of all circRNAs found in the hypothalamus (our result), and only 886 circRNAs were identified in sheep muscle [29]. The abundance of circRNAs detected in the hypothalamus indicates their important roles in hypothalamic functions, such as modulating GnRH pulsatile release. The findings also suggest that some circRNAs may be specific to particular tissues or physiological stages.

### 4.1. Functional Enrichment Analysis of Key circRNAs

CircRNAs have been revealed to act at the promoter region of host genes to enhance their transcription by interacting with U1 small nuclear ribonucleoproteins (snRNPs) and RNA polymerase II [30]. Therefore, determining the potential functions of host genes associated with circRNAs may help in understanding circRNA functions. The TGF-β signaling pathway in ovary was found to modulate reproduction [31,32], but little is known about its effects on hypothalamic functions. Our results suggest that the TGF-β signaling pathway was also enriched in the hypothalamus, suggesting its potential roles in hypothalamic functions. SMAD family member 2 (*SMAD2*), a key member of the SMAD family, was shown to be a host gene of oar_circ_0018794 and particularly associated with the TGF-β signaling pathway. Moreover, SMAD2 was found to be highly enriched in rat hypothalamus [33] and to play key roles in maintaining neuronal differentiation in both human [34] and mouse [35]. In addition, SMAD2 can act as a mediator of TGF-β1 signaling, affecting GnRH gene expression directly in rats [36], indicating its importance in hypothalamic function. Another pathway, the Mitogen-activated protein kinase (MAPK) signaling pathway, was reported to modulate female reproduction, especially in the hypothalamus [37]. Fibroblast growth factor 2 (*FGF2*), which is a source gene of oar_circ_0008291 and enriched in the MAPK signaling pathway, is a critical neurotrophic factor and mitogen for hypothalamic cells in vitro, and its expression decreases at the beginning of puberty in female rats [38], moreover, the production of *FGF2* significantly increased nearly twofold after fasting in rats [39]. Interestingly, a similar acute decrease in leptin was also observed after fasting [25]. Considering the modulatory effects of leptin on GnRH release [40], we speculate that *FGF2* may cooperate with leptin to mediate GnRH activities, although the detailed mechanisms involved require further validation.

Regarding PL vs. ML, the most enriched pathway was endocytosis, which mainly mediates the responses to receptor activity [41]. ArfGAP with RhoGAP domain, ankyrin repeat and PH domain 2 (*ARAP2*), which was enriched in pathway of endocytosis and was a host gene of oar_circ_0034134, was found to reduce glucose uptake and affect sphingolipid metabolism after *ARAP2* knockdown [42]. In addition, high glucose concentration in neuronal cells was shown to influence GnRH activities, impair cell viability, and further result in the apoptosis of GnRH-secreting neuronal cells [43]. It should also be noted that the sphingolipid whose metabolism was affected by *ARAP2* knockdown also mediated the cytoplasmic signaling of estrogens [44], which may be involved in the negative feedback control of estrogen on GnRH secretion in the luteal phase. Thus, ARAP2 may be involved in GnRH regulation by modulating the glucose concentration and negative feedback control of estrogen on GnRH secretion. Other host genes include eukaryotic translation initiation factor 5 (*EIF5*) and ATR serine/threonine kinase (*ATR*), both of which generate the circRNAs oar_circ_0010234 and oar_circ_0013788. *EIF5* was reported to promote GTP hydrolysis and translation initiation complex assembly in eukaryotic translation initiation [45,46], and it also improved the accuracy of start codon recognition [47]. *ATR* could be a sensor to respond to DNA damage [48] and be recruited to the sites of such damage [49]. Additionally, the disruption of *ATR* results in early embryonic lethality in mice [50]. Therefore, the expression of two genes, *EIF5* and *ATR*, may be enhanced by circRNAs to promote the expression of some other genes related to hormone activities such as GnRH release.

### 4.2. ceRNA Analysis Involving oar_circ_0012110

The competitive endogenous RNA network, as an emerging strategy, has been widely used to understand complex processes such as cancer and reproduction [51,52]. In our constructed ceRNA network, oar_circ_0012110 was shown to be a sponge for oar-miR-665-3p, which also targets ten genes. Cooperating with RhoA, diaphanous-related formin 1 (*DIAPH1*) was also reported to mediate the biosynthesis of cortisol by modulating mitochondrial trafficking [53]. In addition, cortisol was also demonstrated to have a key role in reducing the pulse frequency of GnRH on sheep in the follicular phase in the presence of ovarian steroids [54], suggesting the key roles of oar_circ_0012110 in regulating the expression of *DIAPH1*. Further experiments may confirm its functions in mediating GnRH pulsatile secretion.

## 5. Conclusions

In this study, we established the first circRNA expression profile in sheep hypothalamus. We also identified several key circRNAs, such as oar_circ_0018794, oar_circ_0008291, oar_circ_0015119, oar_circ_0012801, circRNAs-oar_circ_0010234, and oar_circ_0013788, through functional enrichment analysis and oar_circ_0012110 through a ceRNA network. Most of these circRNAs may function by influencing GnRH activities or altering key gene expression indirectly or directly. This study presents an integral circRNA analysis in sheep hypothalamus and provides a reference for understanding sheep prolificacy.

## Figures and Tables

**Figure 1 animals-09-00557-f001:**
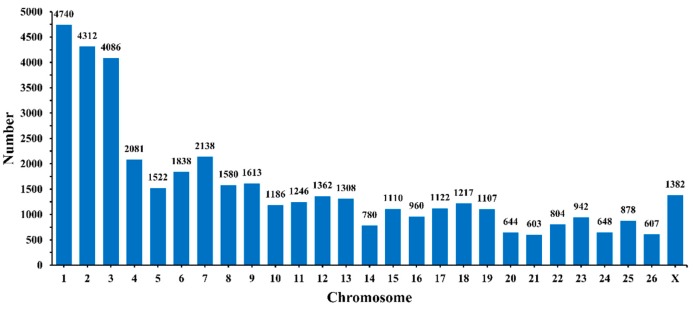
Chromosome distribution of circular RNAs identified from the hypothalami.

**Figure 2 animals-09-00557-f002:**
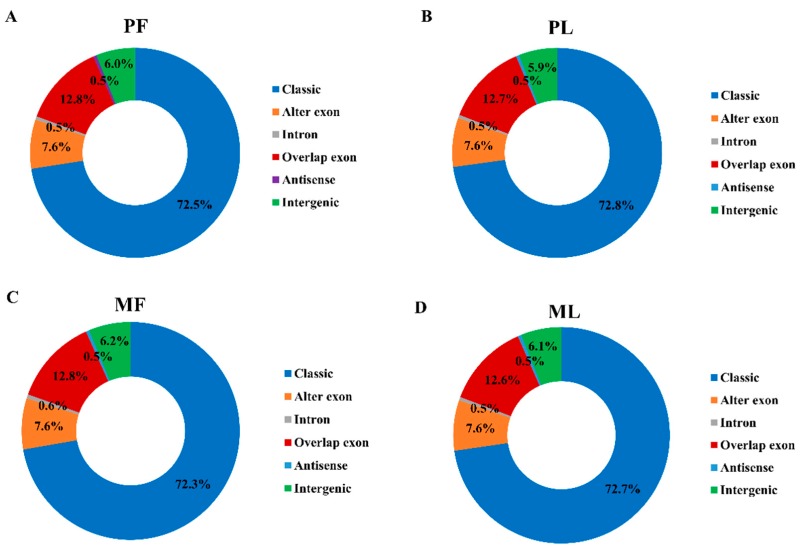
Types of identified circular RNAs. (**A**) The types of circular RNAs identified in polytocous sheep in the follicular phase (**B**), polytocous sheep in the luteal phase, monotocous sheep in the follicular phase (MF) (**C**), and monotocous sheep in the luteal phase (ML) (**D**).

**Figure 3 animals-09-00557-f003:**
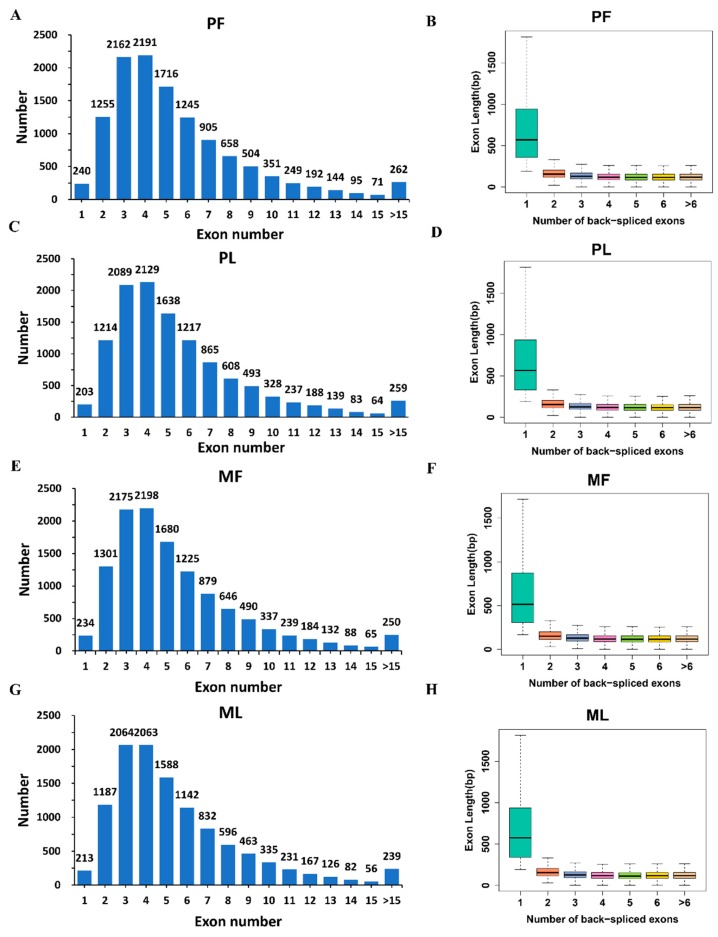
An overview of the exon number and length of identified circular RNAs (circRNAs). The exon number (**A**) and length (**B**) distributions of circRNAs in polytocous sheep in the follicular phase (PF). The exon number (**C**) and length (**D**) distributions of circRNAs in polytocous sheep in the luteal phase (PL). The exon number (**E**) and length (**F**) distributions of circRNAs in monotocous sheep in the follicular phase (MF). The exon number (**G**) and length (**H**) distributions of circRNAs in monotocous sheep in the luteal phase (ML).

**Figure 4 animals-09-00557-f004:**
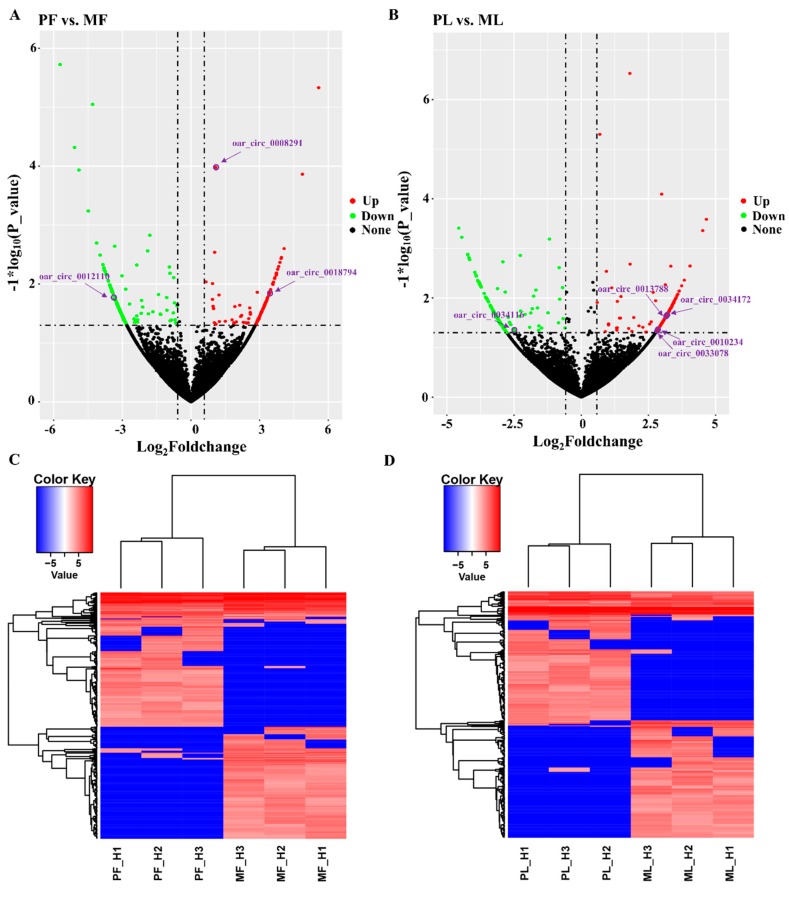
An overview of differentially expressed circular RNAs (DE circRNAs). (**A**) A volcanic plot in polytocous sheep in the follicular phase versus monotocous sheep in the follicular phase (PF vs. MF), in which several key circRNAs are also labeled. (**B**) A volcanic plot in polytocous sheep in the luteal phase versus monotocous sheep in luteal sheep (PL vs. ML), in which several key circRNAs are also labeled. (**C**) The expression pattern of DE circRNAs and hierarchical clustering analysis in PF vs. MF. (**D**) The expression pattern of DE circRNAs and hierarchical clustering analysis in PL vs. ML.

**Figure 5 animals-09-00557-f005:**
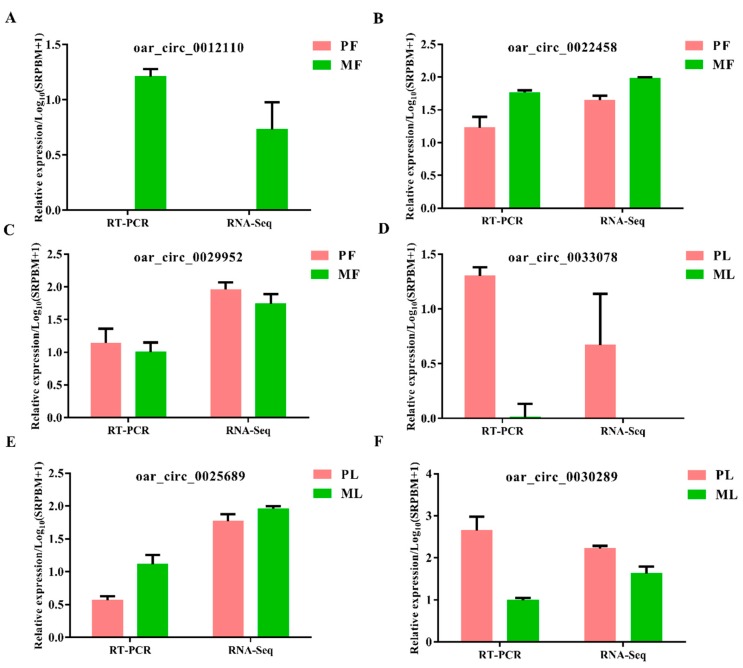
Validation results of three selected circular RNAs in polytocous sheep in the follicular phase (PF) and polytocous sheep in the luteal phase (PL) (**A**–**C**); Validation results of three circular RNAs in monotocous sheep in the follicular phase (MF) and monotocous sheep in the luteal phase (ML) (**D**–**F**) by real-time quantitative polymerase chain reaction (RT-qPCR).

**Figure 6 animals-09-00557-f006:**
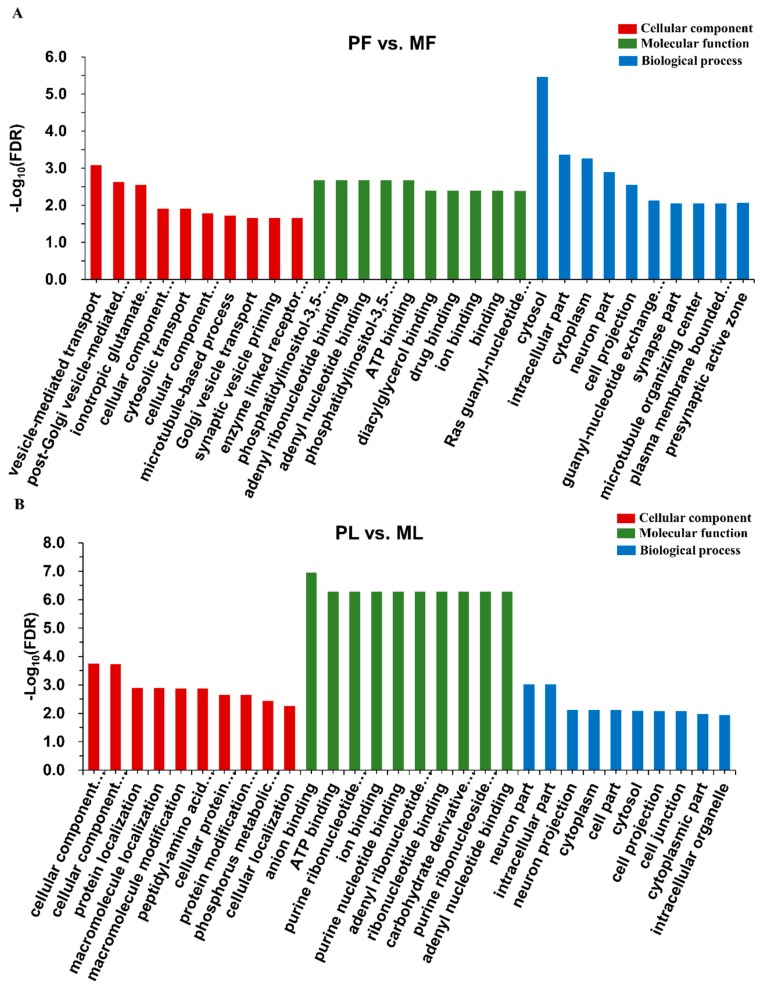
Gene ontology functional enrichment analysis. (**A**) The top ten enriched GO terms in biological process, molecular function, and cellular component, respectively, in polytocous sheep in the follicular phase versus monotocous sheep in the follicular phase (PF vs. MF). (**B**) The top ten GO terms enriched in biological process, molecular function, and cellular component, respectively, in polytocous sheep in the luteal phase versus monotocous sheep in the luteal phase (PL vs. ML). Notes: FDR: False discovery rate.

**Figure 7 animals-09-00557-f007:**
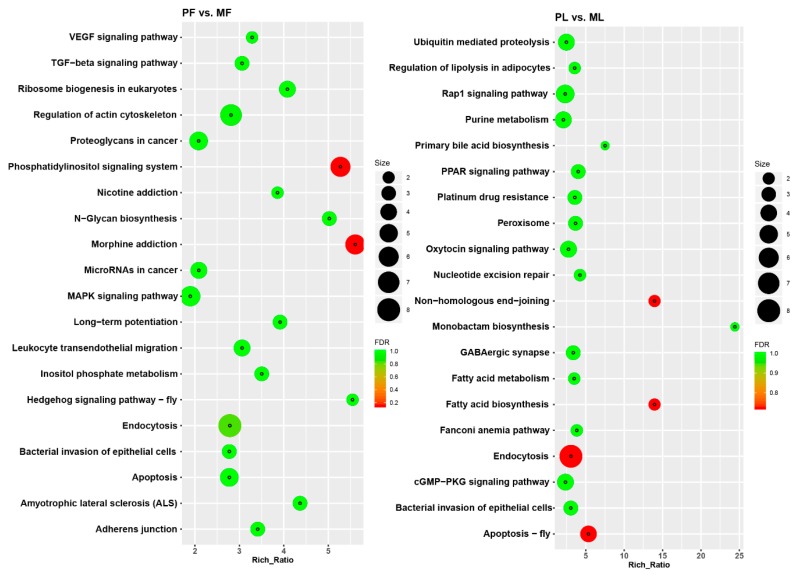
Kyoto Encyclopedia of Genes and Genomes pathway enrichment analysis. (**A**) The top 20 pathways enriched in polytocous sheep in the follicular phase versus monotocous sheep in the follicular phase (PF vs. MF). (**B**) The top 20 pathways enriched in polytocous sheep in the luteal phase versus monotocous sheep in the luteal phase (PL vs. ML). Notes: FDR: False discovery rate.

**Figure 8 animals-09-00557-f008:**
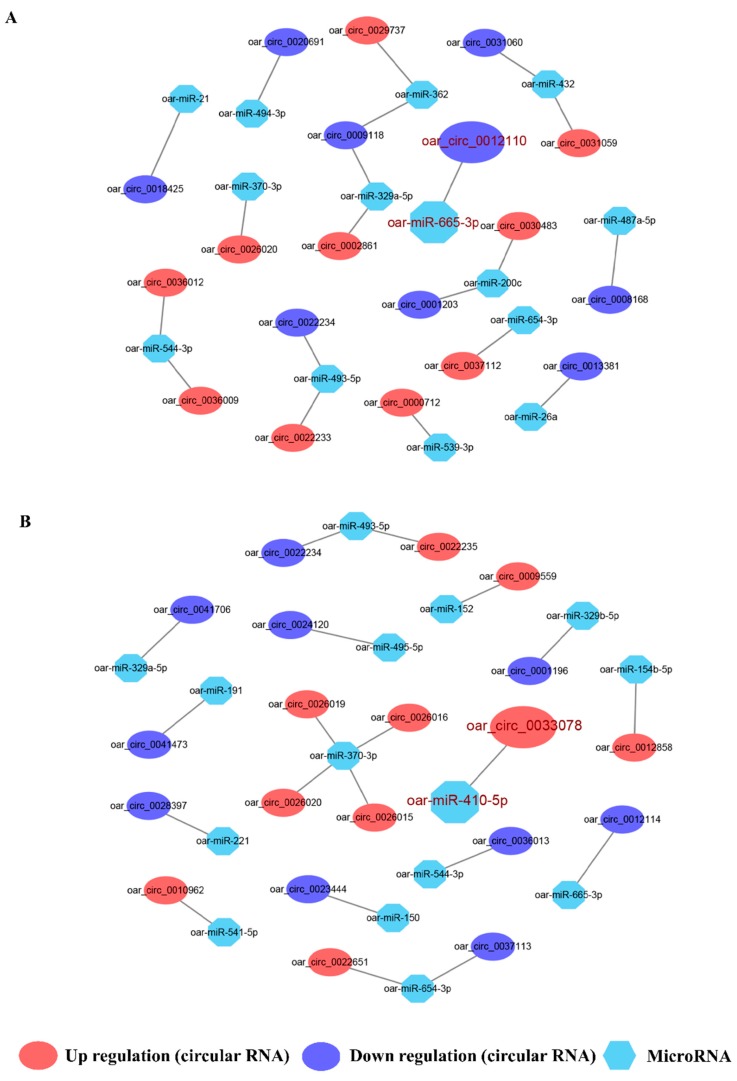
Prediction of microRNA-circular RNA interactive networks. (**A**) The top 20 differentially expressed circular RNAs (DE circRNAs) including the top 10 up- and 10 down-DE circRNAs in polytocous sheep in the follicular phase versus monotocous sheep in the follicular phase (PF vs. MF), in which a key circRNA-miRNA pair was highlighted in red. (**B**) The top 20 differentially expressed circular RNAs (DE circRNAs) including the top 10 up- and 10 down- DE circRNAs in polytocous sheep in the luteal phase versus monotocous sheep in the luteal phase (PL vs. ML), in which a key circRNA-miRNA pair was highlighted in red.

**Figure 9 animals-09-00557-f009:**
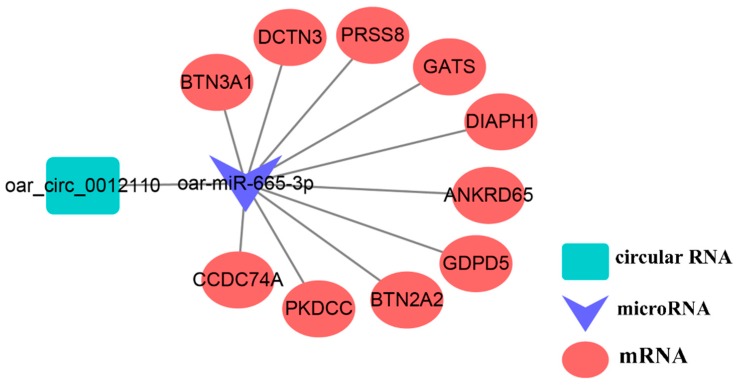
Predicted competing endogenous RNA (ceRNA) network involving oar_circ_0012110.

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
