# Peer review of "Comparative Transcriptomics Identify Key Hypothalamic Circular RNAs that Participate in Sheep (Ovis aries) Reproduction"

_animals, 2019, doi:10.3390/ani9080557_

Round 1
Reviewer 1 Report
animals-563177 . Comparative Transcriptomics Identifies Key Hypothalamic Circular RNAs that Participate in 4 Sheep (Ovis aries) Reproduction
The manuscript is considering about hypothalamic circRNAs that participate in sheep (Ovis aries) reproduction. The authors analyze the expression of circRNA in hypothalamus depending on ovary phase. The theme is highly interesting because concerns the epigenetic regulation in ovine reproduction. Moreover, the circRNA regulation role has not yet been well examined. Maybe, I`m not a good judge in the correction of English language because it is not my native language, but a few sentences should be corrected.
Below a few additional comments:
Line 121 how many cycles were applied during RNA sequencing and they were single or pair-end? By using which chemistry?
Line 130 to which version of genome assembly reads was mapped?
Line 140 |log2Foldchange|> 0.58 please given in FC value which shows the differences in gene expression between analysed groups.
Line 140 You should include the corrected pvalue due to multiply analysis
Line 143 in Go and KEGG analysis also correction for p-value should be applied.
Line 161-165 it is likely the fragment of Animals journal template. Please check. The data should be deposited in to the publicly available database.
Figure 5 is not well described. Moreover, error bars are missing in RNA-seq (Analysis included 3 individuals per group? Or not?)
Result: PCA plot can be added to show the clustering between groups
Line 277 -280; Please rearranged the sentence. In this form is not exactly clear. Moreover please give the citation for pituitary study.
Line 287 "modulate reproduction in ovary"? Maybe: the TGF-b signalling in ovary was found to modulate reproduction
Line 296; the same remark as above: to modulate female reproduction, especially in the hypothalamus? Please rearrange
Availability of data and material – please add the data into public database.
Author Response
Response to Reviewer 1:
animals-563177. Comparative Transcriptomics Identifies Key Hypothalamic Circular RNAs that Participate in Sheep (Ovis aries) Reproduction
The manuscript is considering about hypothalamic circRNAs that participate in sheep (Ovis aries) reproduction. The authors analyze the expression of circRNA in hypothalamus depending on ovary phase. The theme is highly interesting because concerns the epigenetic regulation in ovine reproduction. Moreover, the circRNA regulation role has not yet been well examined. Maybe, I`m not a good judge in the correction of English language because it is not my native language, but a few sentences should be corrected.
Response: We thank you very much for your positive comments on our manuscript.
Below a few additional comments:
Line 121 how many cycles were applied during RNA sequencing and they were single or pair-end? By using which chemistry?
Response: Thanks for your comments. In our research, 150 cycles were applied in RNA sequencing, and they were pair-end. As shown in manuscript, samples of 3 μg of total RNA from each hypothalamus were pooled to construct the RNA library. First, ribosomal RNA was removed using the Ribo-Zero™ Gold Kit (Epicentre, Madison, WI, USA). Then, fragmentation buffer was added to break the RNA into fragments, which were used as templates to synthesize the first strand of cDNA. The second strand of cDNA was also synthesized in the presence of dNTPs, RNase H, and DNA Polymerase I. The obtained double-stranded cDNA was processed with end-repair, the addition of base A and sequencing adaptors, and UNG enzyme digestion. Subsequently, the polymerase chain reaction was performed to construct an RNA library. Finally, the desired RNA fragments were selected for sequencing (Illumina HiSeq 2500), and 150 bp paired-end reads were generated.
Line 130 to which version of genome assembly reads was mapped?
Response: Thanks for your comment. The version of genome assembly was Oar_3.1.
Line 140 |log2Foldchange|> 0.58 please given in FC value which shows the differences in gene expression between analysed groups.
Response: Thanks for your comment. Foldchange > 1.5 has been added in the manuscript.
Line 140 You should include the corrected pvalue due to multiply analysis
Response: Thanks for your comment. About the DE circRNAs, we set thresholds as Foldchange > 1.5 and p < 0.05, which was based on the fact that all the sheep used in this study have the similar genetic background: they were the same breed (Small Tail Han sheep); and all sheep were from the main core production area of Small Tail Han Sheep in Southwest Shandong province, China; all sheep were without the effects of FecB mutation. And there were no significant differences in sheep age, weight, height, body length, chest circumference and tube circumference (P > 0.05), but there were significant differences in ovulation rate or litter size (P < 0.01). It means we had tried to minimize the external and internal factors influencing our experiments; the main difference of these sheep was that they had different fecundity phenotype.
In addition, the design of this research does not expect to narrow down possibility to stringent level, which may ignore some key information, therefore, we choose p < 0.05, which was relatively loose, and our future experiment will validate their real functions.
Therefore, in our results, it is reasonable to take Foldchange > 1.5 and p < 0.05 as a threshold to identify DE circRNAs.
Line 143 in GO and KEGG analysis also correction for p-value should be applied.
Response: Thanks for your comment. About the GO and KEGG analysis, we set thresholds of p < 0.05 as significant level consistent with the standard of DE circRNAs, which was based on the fact that all the sheep used in this study had the similar genetic background (see the response 4).
Therefore, in our results, it is reasonable to take p < 0.05 as a threshold to identify significantly enrichened GO terms and KEGG pathways.
Line 161-165 it is likely the fragment of Animals journal template. Please check. The data should be deposited in to the publicly available database.
Response: Thanks for your comment. We have checked and revised it. About the data deposition, all the RNA-sequencing data used in this study have been deposited in the Sequence Read Archive (SRA) public databases (https://submit.ncbi.nlm.nih.gov/) under BioProject (PRJNA529384).
Figure 5 is not well described. Moreover, error bars are missing in RNA-seq (Analysis included 3 individuals per group? Or not?)
Response: Thanks for your nice suggestion. The description of Figure 5 has been revised in the manuscript. Our analysis included 3 individuals per group, however, about the error bars, we think it was unnecessary, because all the functional analysis was based on the normalized values, therefore, it’s much better to use it to represent expression level, furthermore, there are many previously published paper did it like our (Li et al., 2018; Guo et al., 2018).
Li C Y, Li X, Liu Z, et al. Identification and characterization of long non-coding RNA in prenatal and postnatal skeletal muscle of sheep[J]. Genomics, 2018, 11(2), 133-141. Guo Y Z, Dao G C, Tong Z, et al. Transcriptomic and functional analyses unveil the role of long non-coding RNAs in anthocyanin biosynthesis during sea buckthorn fruit ripening[J]. DNA Research, 2018. doi: 10.1093/dnares/dsy017Result: PCA plot can be added to show the clustering between groups
Response: Thanks for your nice suggestion. We think a heat map clustering (see Figure 4(C, D)) was much better than PCA plot, and hierarchical clustering analysis suggested that four divided groups with each group contains three individuals in our research were relatively rational.
Line 277 -280; Please rearranged the sentence. In this form is not exactly clear. Moreover, please give the citation for pituitary study.
Response: Thanks for your nice suggestion. All of those have been revised in the manuscript.
Line 287 "modulate reproduction in ovary"? Maybe: the TGF-b signaling in ovary was found to modulate reproduction
Response: Thanks for your nice suggestion. It has been revised in the manuscript.
Line 296; the same remark as above: to modulate female reproduction, especially in the hypothalamus? Please rearrange
Response: Thanks for your nice suggestion. It has been revised in the manuscript.
Availability of data and material – please add the data into public database.
Response: Thanks for your nice suggestion. All the RNA-sequencing data used in this study have been deposited in Sequence Read Archive (SRA) public databases (https://submit.ncbi.nlm.nih.gov/) under BioProject (PRJNA529384).
Reviewer 2 Report
Manuscript by Zhuangbiao and colleagues entitled 'Comparative Transcriptomics Identifies Key Hypothalamic Circular RNAs that Participate in Sheep (Ovis aries) Reproduction' aims to explore the expression of key circRNAs in the hypothalamus of sheep by RNA sequencing. Authors revealed several key hypothalamic circRNAs which may participate in sheep reproduction by influencing GnRH activities or affecting expression of key gene, which provided a further insight in sheep fecundity.
Data presented in the manuscript is definitely interesting, but some problems in the manuscript still deserve attention.
Specific comments divided by sections:
Methods:
- The TaqMan MGB probe method was used to genotype the sheep with FecB ++. But the related result of genotype is not shown in this manuscript.
Results:
- As for Figure5, except for “oar-circ-0012110” and “oar-circ-0033078”, the remaining 4 identified circ-RNAs are not reflected in the volcano map (Figure4). Why? It’s better to mark them in the volcano map.
- Figure 8 and Figure 9 are not mentioned in the entire article. Why?
- Please check the reference labels for graphs and tables in “3.3 part”.
Minor comments:
- Line93-97, please check the sentence grammar.
- Line192, according to the figure and sentence meaning, it should be “ML” rather than “PL”.
- Line205, in Figure 4C, it should be “PF”and “MF” instead of “PF”and “PL”.
- Please carefully check all the legends and titles of all figures in the manuscript, especially the abbreviation.
Author Response
Response to Reviewer 2:
Manuscript by Zhuangbiao and colleagues entitled 'Comparative Transcriptomics Identifies Key Hypothalamic Circular RNAs that Participate in Sheep (Ovis aries) Reproduction' aims to explore the expression of key circRNAs in the hypothalamus of sheep by RNA sequencing. Authors revealed several key hypothalamic circRNAs which may participate in sheep reproduction by influencing GnRH activities or affecting expression of key gene, which provided a further insight in sheep fecundity.
Data presented in the manuscript are definitely interesting, but some problems in the manuscript still deserve attention.
Response: We thank you very much for your positive comments on our manuscript.
Specific comments divided by sections:
Methods:
- The TaqMan MGB probe method was used to genotype the sheep with FecB ++. But the related result of genotype is not shown in this manuscript.
Response: Thanks for your nice suggestion. The genotyping results have displayed below, and we selected monotonous and polytocous sheep based on this genotyping result.
Table 1. The genotyping results of 890 Small Tail Han sheep
|
Locus |
Genotype |
Number |
Genotype frequency |
|
|
++ |
142 |
0.16 |
|
FecB |
B+ |
413 |
0.46 |
|
|
BB |
335 |
0.38 |
Results:
- As for Figure 5, except for “oar-circ-0012110” and “oar-circ-0033078”, the remaining 4 identified circ-RNAs are not reflected in the volcano map (Figure 4). Why? It’s better to mark them in the volcano map.
Response: Thanks for your comments. All the circRNAs marked in the volcano map (Figure 4) were regarded as key cicRNAs presuming participating in sheep reproduction, which have been discussed in discussion part, other circRNAs, though as DE circRNAs, may not function in sheep reproduction obviously, therefore we think the circRNAs reflected in volcano map (Figure 4) was reasonable.
- Figure 8 and Figure 9 are not mentioned in the entire article. Why?
Response: Thanks for your carefulness. All those two figures have been added in Line 251 (Figure 8) and Line 258 (Figure 9).
- Please check the reference labels for graphs and tables in “3.3 part”.
Response: Thanks for your carefulness. It has been checked carefully.
Minor comments:
- Line93-97, please check the sentence grammar.
Response: Thanks for your carefulness. It has been checked carefully.
- Line192, according to the figure and sentence meaning, it should be “ML” rather than “PL”.
Response: Thanks for your carefulness. It has been revised carefully.
- Line205, in Figure 4C, it should be “PF” and “MF” instead of “PF” and “PL”.
Response: Thanks for your carefulness. It has been checked carefully.
- Please carefully check all the legends and titles of all figures in the manuscript, especially the abbreviation.
Response: Thanks for your carefulness. All the legends and titles of all figures in the manuscript have been checked carefully.
Round 2
Reviewer 1 Report
The manuscript is considering about hypothalamic circRNAs that participate in sheep (Ovis aries) reproduction. The authors analyze the expression of circRNA in hypothalamus depending on ovary phase. The theme is highly interesting because of concerns the epigenetic regulation in ovine reproduction. Moreover, the circRNA regulation role has not yet been well examined. The authors included almost all reviewer suggestions.
Figure 5 - if the authors presented the mean between 3 individuals should give the error of SD bars, it does not matter what kind of results there are, RNA-seq or qPCR.
Moreover, including the adjusted P-value in Deseq analysis is due to the multiply analysis between numerous genes, does not matter if the sheeps had a similar genetic background. the same in functional analysis (KEGG Panther).
Information about SRA files and bioproject (PRJNA529384) should be added into the manuscript.
The English language was only slightly corrected so the authors should ask a native speaker for help.
Author Response
Response to Reviewer 1
The manuscript is considering about hypothalamic circRNAs that participate in sheep (Ovis aries) reproduction. The authors analyze the expression of circRNA in hypothalamus depending on ovary phase. The theme is highly interesting because of concerns the epigenetic regulation in ovine reproduction. Moreover, the circRNA regulation role has not yet been well examined. The authors included almost all reviewer suggestions.
Figure 5 - if the authors presented the mean between 3 individuals should give the error of SD bars, it does not matter what kind of results there are, RNA-seq or qPCR.
Response: Thanks for your comment, the SD bars has been added in the RNA-seq results.
Moreover, including the adjusted P-value in Deseq analysis is due to the multiply analysis between numerous genes, does not matter if the sheep had a similar genetic background. the same in functional analysis (KEGG Panther).Response: Thanks for your comments, the adjusted P-value (q value) of Genes have been added in supplemental file (Table S4), and adjusted P-value (FDR) has been added in GO and KEGG analysis (Figure 6 and 7).
Information about SRA files and bioproject (PRJNA529384) should be added into the manuscript.Response: Thanks for your comment, it has been added in the manuscript.
The English language was only slightly corrected so the authors should ask a native speaker for help.Response: Thanks for your comment, the manuscript has been corrected by native speaker (see attach file).

Reviewer 2 Report
Minor comments:
In the modified version, all the figures are distorted or ambiguous. Please check or replace them.
Author Response
Response to Reviewer 2
In the modified version, all the figures are distorted or ambiguous. Please check or replace them.
Response: Thanks for your comment, all the figures has replaced with clear figure (300dpi), we also uploaded all figures respectively.